# Type II Grass Carp Reovirus Infects Leukocytes but Not Erythrocytes and Thrombocytes in Grass Carp (*Ctenopharyngodon idella*)

**DOI:** 10.3390/v13050870

**Published:** 2021-05-10

**Authors:** Ling Yang, Jianguo Su

**Affiliations:** 1Department of Aquatic Animal Medicine, College of Fisheries, Huazhong Agricultural University, Wuhan 430070, China; Yangling201807@163.com; 2Laboratory for Marine Biology and Biotechnology, Pilot National Laboratory for Marine Science and Technology (Qingdao), Qingdao 266237, China; 3Engineering Research Center of Green Development for Conventional Aquatic Biological Industry in the Yangtze River Economic Belt, Ministry of Education, Wuhan 430070, China

**Keywords:** grass carp (*Ctenopharyngodon idella*), grass carp reovirus (GCRV), leukocytes, erythrocytes, thrombocytes, virus inclusion bodies (VIBs), *C. idella* kidney (CIK) cells

## Abstract

Grass carp reovirus (GCRV) causes serious losses to the grass carp industry. At present, infectious tissues of GCRV have been studied, but target cells remain unclear. In this study, peripheral blood cells were isolated, cultured, and infected with GCRV. Using quantitative real-time polymerase chain reaction (qRT-PCR), Western Blot, indirect immunofluorescence, flow cytometry, and transmission electron microscopy observation, a model of GCRV infected blood cells in vitro was established. The experimental results showed GCRV could be detectable in leukocytes only, while erythrocytes and thrombocytes could not. The virus particles in leukocytes are wrapped by empty membrane vesicles that resemble phagocytic vesicles. The empty membrane vesicles of leukocytes are different from virus inclusion bodies in *C. idella* kidney (CIK) cells. Meanwhile, the expression levels of *IFN1*, *IL-1β*, *Mx2*, *TNFα* were significantly up-regulated in leukocytes, indicating that GCRV could cause the production of the related immune responses. Therefore, GCRV can infect leukocytes in vitro, but not infect erythrocytes and thrombocytes. Leukocytes are target cells in blood cells of GCRV infections. This study lays a theoretical foundation for the study of the GCRV infection mechanism and anti-GCRV immunity.

## 1. Introduction

Aquaculture is a major global industry with a total annual production exceeding 179 million tons and an estimated value of almost 401 billion US dollars (FAO, 2018). Among them, grass carp output reached 5.533 million tons, accounting for 18.36% of freshwater aquaculture output. However, the severe hemorrhagic disease caused by grass carp reovirus (GCRV) in grass carp fingerlings every year, with a mortality rate of up to 90%, has caused significant losses to the aquaculture industry [1,2,3]. GCRV particles are icosahedral with a double capsid structure that contains a genome of eleven double-stranded RNA (dsRNA) segments encoding 13 proteins, including 7 structural and 6 non-structural proteins [4,5]. GCRV is classified into three types, represented by strains GCRV-873 (type I), GCRV-GD108 (type II), and GCRV104 (type III) [6,7]. Phylogenetically, all three GCRV types are tentatively placed into the *Aquareovirus* genus whose members each contain 11 segments, nevertheless, GCRV clearly shows a divergence from the approved species in this genus. In fact, GCRV’s appear to share similarities with a salmon virus is known as piscine orthoreovirus (PRV) placed within the *orthoreovirus* genus whose members contain 10 segments [8].

*Orthoreoviruses* enter target cells either through a receptor-mediated mechanism or through a combined process of extracellular outer capsid proteolysis and receptor-independent uptake [9]. PRV replicates in the cytoplasm [10]. The erythrocyte is the primary cell type targeted by PRV in salmon, and PRV-1 protein and genetic material amplification both occur within erythrocytes [11]. The liver and head kidney were heavily infected by PRV in Atlantic salmon, and hemolysis of blood cells has been observed in the spleen and head kidney [12]. Reported target cells for PRV in Atlantic salmon are erythrocytes, myocytes, and macrophages [13,14,15,16]. In the acute phase of infection, up to 50% of the erythrocytes contain many PRV virus inclusions in the cytoplasm [15,17], PRV is also released to high levels in plasma in this phase [15,18]. If virus release occurs through cell-lysis is unclear. The *C. idella* kidney (CIK) cells are sensitive to GCRV, and GCRV can be proliferated and cultured in CIK cells. Many studies have been conducted on the cytopathological characteristics of GCRV by using CIK cells as the material basis, and different genotypes of GCRV cause different cytopathological changes [19]. In grass carp, GCRV leads to massive abdominal hemolysis and apparent hemorrhage in muscle, skin, intestine, and gill [19]. However, the main target organ of GCRV infection in grass carp is still controversial. Studies have shown that the intestine’s surface has many functional receptors for GCRV, it has been speculated that is the main target organ for GCRV infection. GCRV penetrates the blood and lymph tissue through the intestine to infect other organs [20]. Studies have also shown that GCRV enters the host fish mainly through fish gills [21], and the difference in the main target organs may be caused by the different ways the virus infects the fish [22]. At the same time, studies have shown that GCRV infection of grass carp will cause significant changes in the blood parameters of the fish body, and cause the vasodilation and congestion of the organs, the infiltration of lymphocytes and macrophages, and the severe vacuolar degeneration of the spleen, kidney, and liver [23]. The disease is mainly reflected in the destruction of the blood circulatory system and parenchymal organs [24], but there are still gaps in the blood cell types of GCRV infected grass carp. In the antiviral response of grass carp, *IFN1* and *NF-κB* are not direct effectors in the antiviral and inflammatory responses, and their downstream effectors can eventually play antiviral and pro-inflammatory roles. The downstream effector genes of *NF-κB* are mainly *TNFα* and *IL-1β*, which reflect the changes of inflammation [19]. The downstream effector genes of *IFN1* mainly include *Mx*, *Viperin*, *Gig* and so on [25]. Among them, *Mx2* can inhibit the further proliferation and replication of GCRV in cells, reflecting the natural antiviral immune response [26].

A substantial amount of literature has reported on the types and structural characteristics of fish blood cells [27]. Different cell type has been defined not only structurally but also functionally. It is generally accepted that fish blood cells follow the basic hematological pattern of higher vertebrates and constitute erythrocytes, thrombocytes, and leucocytes, leucocytes can be subdivided into lymphocytes, monocytes, and granulocytes [28]. Erythrocytes in fish blood account for more than 90% of the total blood cells [29]. The main function of erythrocytes in fish is to carry and transport oxygen and carbon dioxide. Simultaneously, it can also maintain the ion balance of the internal environment by buffering some acid-base substances produced by metabolism in the organism. Studies have found that erythrocytes can be used as immune cells and have certain phagocytic functions, so they also exhibit certain immune functions [30]. As the basis of immune defense, lymphocytes account for about 80% of the white blood cells of fish, and their main function is to produce antibodies [29]. Monocytes have active deformation movement, obvious chemotaxis, and certain phagocytic function [31]. Research results in recent years have shown that thrombocytes also have phagocytic functions and participate in non-specific and specific immune responses [32]. After GCRV infection, the number of lymphocytes and monocytes increased, while the number of neutrophils decreased. These three kinds of cell are the most important leukocytes in grass carp. At the same time, it can also cause the decrease of the number of erythrocytes, the decrease of hemoglobin, the decrease of the number of thrombocytes and the increase of the platelets volume, which are consistent with the typical bleeding symptoms of grass carp hemorrhage [23].

Some studies have shown that the GCRV virus enters cells through cell endocytosis [33]. The way GCRV enters the cell may be related to the endocytosis mediated by clathrin and caveolin. Some studies have shown that the adhesion protein σ1 of *orthoreovirus* uses the transmembrane protein junctional adhesion molecule A (JAMA) on the surface of the host cell as the receptor and enters the cell through the ligand-receptor pattern recognition pathway, thereby achieving the purpose of infecting the cell [34,35]. The outer capsid protein VP56 of GCRV has the highest homology with the adhesion protein σ1 protein of *orthoreovirus*, and it is likely to play an adhesion role. Therefore, GCRV particles invade cells by the adhesion of VP56 protein to the cell surface. However, some studies have found that JAMA protein cannot interact with the outer capsid protein of GCRV. The JAMA protein used by *orthoreovirus* cannot be used as a receptor for GCRV to enter the cell. However, no effective receptor for GCRV has been found so far. Some researchers speculate that GCRV enters cells through different pathways. At present, the way GCRV enters cells is still controversial.

For a long time, the target organs and tissues of GCRV infection of grass carp have been studied in-depth, but the target cells of GCRV infection of grass carp are rarely studied. Studies have shown that GCRV can enter cells through cell endocytosis. The way that GCRV enters the cell may be related to the endocytosis mediated by clathrin and caveolin, but JAMA protein cannot interact with the outer capsid protein of GCRV. Therefore, in the peripheral blood cells, which type of blood cell can be infected by GCRV to perform the function of carrying the virus needs to be studied urgently. In this study, peripheral blood cells were isolated, cultured, and infected separately. Through quantitative real-time polymerase chain reaction (qRT-PCR), Western Blot, indirect immunofluorescence, flow cytometry, transmission electron microscopy, and other methods were used to establish the model of blood cells infected with GCRV in vitro. By exploring the types of blood cells that can be infected by GCRV, the target cell types of grass carp infected by GCRV were further determined. It is concluded that GCRV infects leukocytes but not erythrocytes and thrombocytes of grass carp. It provides a theoretical basis and broadens the field of vision for research on anti-GCRV infection, which is of great significance to the anti-virus research of GCRV.

## 2. Materials and Methods

### 2.1. Ethics Statement

The protocol of animal experiments was approved by the Animal Management and Ethics Committee, Huazhong Agricultural University. The approval number is HZAUFI-2019-021 (Approval date: 21 July 2019). This experiment conscientiously followed the ethical principles of animal welfare. All surgery was performed under MS-222 anesthesia to minimize suffering.

### 2.2. Blood Samples, Isolation of Peripheral Blood Cells of Grass Carp

Grass carp were purchased from a fish farm in Huanggang City (Hubei Province, China), weighing 25 g–30 g, and was temporarily raised in the recirculating freshwater system at 28 ± 1 °C for more than two weeks. They were fed twice a day with a commercial pellet diet at a rate of 2% body weight. 3-aminobenzoic acid ethyl ester methanesulfonate (MS-222) (Sigma-Aldrich Co., St. Louis, MO, USA) was used to anesthetize grass carp by bath immersion for 2–5 min, then the blood samples (0.5 mL) were collected by venipuncture from the caudal with a syringe of 1% heparin wetting. About 5 mL blood samples of 10 grass carp were collected each time, and peripheral blood erythrocytes, leukocytes, and thrombocytes were isolated by used a fish peripheral blood platelets separation kit (Solarbio, Beijing, China).

### 2.3. Cell Culture, Viral Infection, and Antibodies

*Ctenopharyngodon idella* kidney (CIK) cells were provided by the China Center for Type Culture Collection and then expanded by our laboratory. CIK Cells, erythrocytes, thrombocytes, leukocytes were grown in Dulbecco’s modified Eagle’s medium (DMEM, Life Technologies, New York, NY, USA) and L-15 (Leibovitz, Beijing, China) supplemented with 10% FBS (Life Technologies, New York, NY, USA), 100 U/mL penicillin (Sigma, St. Louis, MO, USA) and 100 U/mL streptomycin (Sigma, St. Louis, MO, USA). Cells maintained at 28 °C in a humidified atmosphere of 5% CO_2_ incubator (Thermo Scientific, Waltham, MA, USA). GCRV-II strain GCRV-097 was conserved in our lab. For viral infection, CIK cells were cultured for 24 h in advance and then infected with GCRV-097 at a multiplicity of infection (MOI) of 1 as previously described [36]. The erythrocytes, thrombocytes, leukocytes were adhered and cultured for 4 h, then replaced with a medium containing 3% fetal bovine serum (FBS), and then infected with GCRV-097 at the same dose as the CIK cells. The anti-VP4 mouse polyclonal antibody was prepared and conserved by our lab [37]. Anti-β-actin mouse monoclonal antibody was purchased from Abcam (Cambridgeshire, Britain).

### 2.4. Hemocyte Morphologic Observation and Giemsa stain

The separated leukocytes, thrombocytes, erythrocytes were made into blood smears and observed under a light microscope. The blood smears were fixed in absolute methanol for 3 min at room temperature, and stained with Giemsa (Solarbio, Beijing, China) for 15 min and finally rinsed with distilled water, air dried, and sealed with neutral balsam (Sinopharm Chemical Reagent Co., Ltd. Shanghai, China). The stained smear was photographed under Nikon Microphot FX, and the morphology of leukocytes was observed.

### 2.5. RNA isolation, Polymerase Chain Reaction (PCR) Detection and Western Blotting Analysis

Total RNAs were extracted with TRIZOL (Simgen, Hangzhou, China), and converted to cDNA using the Reverse Transcription Kit HiScript II Q RT SuperMix for qPCR (+gDNA wiper) (Vazyme Biotech Co., Ltd., Nanjing, China).

For Western blotting (WB) analysis, protein extracts were separated by 8% sodium dodecyl sulfate polyacrylamide gel electrophoresis (SDS-PAGE) gels and transferred onto nitrocellulose membranes (Millipore, MA, America). The membranes were blocked in fresh 2% albumin from bovine serum albumin (BSA) dissolved in TBST buffer at 4 °C overnight, then incubated with appropriate indicated primary Abs for 2 h at room temperature. They were then washed three times with TBST buffer and incubated with secondary Ab for 1 h at room temperature. After washing four times with TBST buffer, the nitrocellulose membranes were scanned and imaged by an Odyssey CLx Imaging System (LI-COR) or an Image Quant (GE). The results were obtained from three independent experiments.

### 2.6. Flow Cytometry Analysis

Leukocytes, erythrocytes, and CIK cells after GCRV infection for 0 h, 12 h, 24 h, 36 h, and 48 h were collected into a centrifuge tube and washed in staining buffer (phosphate buffered saline (PBS) + 3% BSA). Before intracellular staining, the cells were fixed in paraformaldehyde for 10 min, and permeabilized by incubation in Triton X-100 for 10 min. Cells were stained with Anti-VP4 (1:500) for 2 h and secondary FITC-conjugated goat anti-mouse IgG (1:50, ABclonal, Wuhan, China) for 1 h. All washing solutions and dilutions are diluted with staining buffer. The cells were read on a Cytoflex S Flow Cytometer (Beckman Coulter, California, America), counting 10,000 cells per sample.

### 2.7. Immunofluorescence Microscopy

The leukocytes, erythrocytes, and CIK cells after GCRV infection for 48 h were fixed with 4% (*v/v*) paraformaldehyde for 5 min, permeabilized with 0.1% (*v/v*) Triton X-100 for 10 min. Subsequently, they were blocked with the non-specific binding with 2% (*v/v*) BSA at 37 °C for 2 h. The slides were washed three times with PBST and then incubated with mouse anti-vp4 antibodies (1:1000) and FITC-conjugated goat anti-mouse IgG (1:200, ABclonal, Wuhan, China) at 37 °C for 2 h and 45 min. The nuclei of all cells were stained with 1 μg/mL Hoechst 33,342 at room temperature for 10 min. Afterward, the stained cells were rinsed with PBS. Images were taken with an UltraVIEW VoX 3D Live Cell Imaging System (PerkinElmer, Waltham, MA, USA).

### 2.8. Transmission Electron Microscope

To observe the virus particles in the cells, leukocytes, erythrocytes, thrombocytes, and CIK cells infected with GCRV for 48 h were fixed with 2.5% glutaraldehyde in 0.1 M phosphate buffer (pH 7.2) for over 24 h at 4 °C. Ultrathin sections were prepared as described previously [38]. Images were viewed on an HT-7700 transmission electron microscope (TEM, Hitachi, Tokyo, Japan).

### 2.9. Quantitative Real-Time PCR Assay

Quantitative real-time PCR (qRT-PCR) was established in a Roche LightCycler^®^ 480 system, and *EF1α* was employed as an internal control gene for cDNA normalization. All the cDNA concentrations were adjusted to 50 ng/µL. The qRT-PCR amplification was carried out in a total volume of 15 µL, containing 7.5 µL of BioEasy Master Mix (SYBR Green) (Hangzhou Bioer Technology Co., Ltd. Hangzhou, China), 3.1 µL of nuclease-free water, 4 µL of diluted cDNA (200 ng), and 0.2 µL of each gene-specific primer (10 µM). The PCR cycling conditions were as follows: 1 cycle of 95 °C for 30 s, 45 cycles of 95 °C for 5 s, 60 °C for 30 s, 1 cycle of 95 °C for 15 s, 60 °C for 30 s, followed by dissociation curve analysis to verify the amplification of a single product. mRNA expression levels were normalized to the expression level of *EF1α*, and the data were analyzed using the 2^−ΔΔCT^ method. The qRT-PCR primers were designed by Primer Premier 5 software based on the gene sequence information in GenBank. They are shown in Table 1.

### 2.10. Statistical Analysis

Statistical analyses and presentation graphics were carried out using the GraphPad Prism 6.0 software (GraphPad Software, San Diego, CA, USA). Results were presented as mean ± standard deviation (SD) for at least three independent experiments. The data were analyzed using an unpaired, two-tailed Student’s *t*-test. *p* Values below 0.05 were regarded as being significant for all analyses (* *p* < 0.05, ** *p* < 0.01).

## 3. Results

### 3.1. Isolation and Identification of Peripheral Blood Cells of Grass Carp

We used the kit to separate the peripheral blood cells of grass carp into red blood cells (RBC), white blood cells (WBC), and thrombocytes (TC) for observation and identification. After centrifugation, the separation liquid was divided into 4 layers. The uppermost layer was thrombocytes, the second layer leukocytes, the third layer the separation liquid layer, and the lowermost layer erythrocytes. Observing the separated erythrocytes in the RBC layer under light microscopy, almost only erythrocytes were present. The WBC layer was rich in cell types, including lymphocytes, neutrophils, monocytes, etc. Of course, there were a tiny amount of erythrocytes and thrombocytes that could not be removed entirely. The most of separated cells in the TC layer were thrombocytes, a very small quantities of erythrocytes cannot be eliminated (Figure 1A). The WBC layer was analyzed by flow cytometry, and the cell type was judged according to the abundance and size of the cells. The results showed that the proportion of lymphocytes in the leukocytes was as high as 83% (Figure 1B). Simultaneously, the leukocytes were analyzed by Giemsa staining, and many lymphocytes were observed, which contain large lymphocytes and small lymphocytes (Figure 1C). The TC layer was analyzed by flow cytometry, and the cell type was judged according to the abundance and size of the cells. The results showed that the proportion of thrombocytes in the TC layer was 57%, and the rest were erythrocytes and cell debris (Figure 1D). At the same time, the thrombocytes were analyzed by Giemsa staining, and it was observed that the thrombocytes had larger nuclei and smaller cytoplasm (Figure 1E).

### 3.2. Grass Carp Reovirus (GCRV) Load in Grass Carp Leukocytes

To determine whether GCRV can infect leukocytes, erythrocytes, and thrombocytes, CIK cells were used as a positive control, and the cells were artificially infected with the same amount of GCRV and PBS for 48 h. PCR results showed that with EF1α as the internal control, only CIK cells and leukocytes after challenge could detect bands, but no bands were detected in erythrocytes, thrombocytes, and uninfected CIK cells and leukocytes after challenge (Figure 2A). WB results revealed that with β-actin as the reference, CIK cell protein had a transparent band at 68 kDa, leukocytes protein had a corresponding band at the same position, and no band was detected in erythrocytes and thrombocytes (Figure 2B). These results indicated that the infected leukocytes and CIK cells can detect the virus’s presence, but erythrocytes and thrombocytes cannot be infected by GCRV. We have used VP4 and VP56 primers to detect three kinds of cells infected with GCRV at 12 h, 24 h, 36 h, and 48 h by qRT-PCR. Only leukocytes showed an increasing trend over time, while erythrocytes and thrombocytes had no significant changes, so erythrocytes and thrombocytes are GCRV-negative. The qRT-PCR analysis of leukocytes at 12 h, 24 h, 36 h, and 48 h after infection display that the mRNA expression of VP4 and VP56 increased logarithmically with the increase of time, indicating that the virus will proliferate in leukocytes over time (Figure 2C,D).

### 3.3. Ex Vivo Infected Leukocytes Trigger an Antiviral Immune Response

To determine whether GCRV-infected leukocytes respond to GCRV infection through antiviral gene expression detection, the levels of *IFN1*, *Mx2*, *IL-1β*, and *TNFα* were analyzed. *IFN1*, *IL-1β*, and *TNFα* were significantly up-regulated at 12 h (Figure 3A,C,D). The expressions of these three genes were similar, they started to be upregulated at 12 h, and the expression levels showed a continuously upward trend. The expression level of *Mx2* began to increase at 24 h, and then offered a constant rising trend (Figure 3B). Simultaneously, the change of *IFN1* expression in erythrocytes did not show apparent regularity with the increase of infection time (Figure 3E). The expression of *IFN1* in thrombocytes was significantly risen at 24 h, and after that it did not show regular changes, but it showed an overall escalating trend (Figure 3F).

### 3.4. GCRV Virus Particle Morphology Observed by Transmission Electron Microscopy (TEM)

TEM analysis of the infected CIK cells, erythrocytes, leukocytes, and thrombocytes showed multiple forms of virus particles and the shape of virus inclusion bodies. CIK cells contained various forms of virus existence, virus particles neatly arranged like a lattice, with a diameter of about 70–80 nm (Figure 4A), and scattered virus particles were dispersed in the cytoplasm and are about 70–80 nm in diameter (Figure 4B). Early virus inclusion bodies contained a small number of virions (Figure 4C), while late virus inclusion bodies were denser, darker and include a large number of virions in the process of formation (Figure 4D). In CIK cells, viral inclusion bodies had dense structures, dark colors, and contained a mass of forming virions. Virus inclusion bodies had different shapes and sizes, but none have apparent membrane structures (Figure 5A,B). However, the virions in leukocytes after the infection were wrapped by distinct membrane structure, and the interior was not dense, showing vacuolation (Figure 5C,D). There were no virus particles in infected erythrocytes, and the cell structure was complete without noticeable pathological changes. Erythrocytes had a long, oval nucleus, which was rich in heterochromatin, the cytoplasm was highly electron-dense and contained few organelles (Figure 5E). There were no virus particles exist in infected thrombocytes, and the cell structure is intact, there were no noticeable pathological changes. Thrombocytes had elongated nuclei rich in heterochromatin, and the length of the cells is about 6–7 μm (Figure 5F).

### 3.5. Viral Inclusions Detected by Indirect Immunofluorescence

GCRV-positive leukocytes were identified by immunofluorescence microscopy by granular staining in the cytoplasm, confocal microscopy of different staining patterns in the cytoplasm, including a few inclusions and scattered granular staining in the perinuclear region (Figure 6A,B). Immunofluorescence microscopy results showed that erythrocytes were GCRV-negative. No fluorescent particles were found in the cytoplasm of the erythrocytes, only the nuclei showed blue fluorescence, indicating that no virus particles were detected in the infected erythrocytes (Figure 6C). Indirect immunofluorescence results showed that CIK cells were GCRV-positive. A mass of virions were detected in the CIK cytoplasm after infection, containing inclusions that varied in size and number from a few large inclusions to multiple more minor inclusions in the perinuclear region, or diffuse granular staining throughout the cytoplasm (Figure 6D).

### 3.6. High Numbers of GCRV-Positive Leukocytes Detected by Flow Cytometry

GCRV-positive leukocytes, GCRV-negative erythrocytes and GCRV-positive CIK cells were detected by flow cytometry. The proportion of positive (green) and negative (blue) GCRV infections was measured by flow cytometry at 0 h, 12 h, 24 h, 36 h, and 48 h after GCRV infection. In ex vivo infected leukocytes, GCRV-positive rates were 0.32%, 23.47%, 33.02%, 45.95%, 64.60% at 0 h, 12 h, 24 h, 36 h, and 48 h, respectively. This result indicated that with the increase of infection time, the number of positive cells also increased, and the overall positive rate of GCRV leukocytes also increased (Figure 7A). In ex vivo infected erythrocytes, GCRV-positive rates were 0.01%, 0.64%, 0.38%, 0.45%, 1.04% at 0 h, 12 h, 24 h, 36 h, and 48 h, respectively. This result indicated that with the duration of infection time, the positive rate of GCRV did not change significantly, indicating that GCRV could not infect erythrocytes in vitro (Figure 7B). In infected CIK cells, GCRV-positive rates were 0.24%, 38.85%, 43.96%, 64.35%, 86.78% at 0 h, 12 h, 24 h, 36 h, and 48 h, respectively. This result indicated that with the increase of infection time, the positive rate of GCRV increased, and the positive rate was all higher than that of leukocytes. It is suggested that GCRV could infect CIK cells, and CIK cells were more likely to infect GCRV than leukocytes (Figure 7C).

## 4. Discussion

In this study, we proved that GCRV could infect leukocytes, but not erythrocytes and thrombocytes. Flow cytometry and qRT-PCR respectively detected a large amount of GCRV protein and RNA in leukocytes. The GCRV protein is localized in the cytoplasm. Indirect immunofluorescence and confocal microscopy observed virus inclusion bodies in the cytoplasm that resembled a virus factory. Transmission electron microscopy showed that these inclusion bodies contained reovirus-like particles.

Immunofluorescence and confocal microscopy observed that the virus-like particle structure exists in the cytoplasm of leukocytes and CIK cells after GCRV infection. In CIK cells, the virus inclusion bodies in the cytoplasm are densely structured, have different shapes, and have no obvious membrane structure wrapping. They are usually located in the perinuclear area. The darker color is observed under the transmission electron microscope, which can be called a “virus factory”. These virus factories contain vast viral proteins and partially or completely assembled particles, which are the location of viral RNA replication, cationic packaging, and assembly of progeny particles [39,40]. In this study, it was observed that the VIBs in CIK cells are the same as the previous studies [41]. The morphology of the VIBs is different. Some inclusion bodies are filled with many virus particles, and almost all of them are mature virus particles, which are defined as “late virus inclusion bodies”. Other inclusion bodies contain a small number of virus particles lighter in color and more uniform in structure, and some do not even contain virus particles, which are called “early virus inclusion bodies”. The morphology of virus particles scattered in the cytoplasm of CIK cells is consistent with that of the virus particles that are being formed in the virus inclusion bodies, indicating that the virus particles in the VIBs are released into the cytoplasm after maturation, which indirectly confirms the virus replication and assembly in the VIBs. In leukocytes, we observed that the virus particles are wrapped by a membrane-like structure. The inner part of this structure is not dense, vacuolar, and the envelope is clear, which is similar to the phagocytic vesicles that gather the virus particles after phagocytosis. This structure is different from the virus inclusion body in CIK cells. However, whether the membrane-like form observed in our study is lipid or protein in nature is still uncertain. In this study, the diameter of reovirus particles observed by transmission electron microscopy was about 70~80 nm, which was in line with the characteristics of the Reoviridae virus particles [42].

GCRV is tentatively placed into the *Aquareovirus* genus yet the distinction between the *aquareovirus* and *orthoreovirus* genus is currently not well defined given the phylogenetic ordination of GCRV and a salmon *orthoreovirus*, PRV [8]. All genotypes of PRV target red blood cells for principal replication which under some circumstances can result in circulatory disease. For instance, specific isolates of PRV-1 have been demonstrated to cause severe heart inflammation [43] while a PRV-2 isolate has been shown to be the aetiological driver of an anaemic condition of farmed Coho Salmon [44]. Interestingly, the presence of many GCRV virus particles in the leukocytes of grass carp is consistent with pathological symptoms such as inflammation and bleeding in the spleen, trunk kidney, head kidney, and other tissues caused by grass carp hemorrhagic disease. Because the spleen and kidney tissues are the most important immune system organs and the main hematopoietic organs, they contain abundant leukocytes [20]. Therefore, the virus infects leukocytes and spreads them to various target organs, resulting in a sizeable viral load in this organ. Simultaneously, these two organs have an important immune function to eliminate circulating cells infected by the virus. Moreover, GCRV can also cause pathological symptoms such as inflammation of the intestines. In histopathological sections, these diseases are manifested as many inflammatory cell infiltrations, and leukocytes are the primary source of inflammatory cells. Therefore, leukocytes are the target cells of GCRV, which are also compatible with the histopathological changes of grass carp hemorrhagic disease.

Blood cells act as the primary executors of cellular immune defense [45], which is a major target cell for Oyster herpesvirus (OsHV-1) and white spot syndrome virus (WSSV) [46,47]. Semigranular cells and granulosa cells are two types of host cells specifically targeted by WSSV in shrimp. In Ark Clam, the target cell of OsHV-1 is type II granulosa cells [46,48]. The granulosa cells of crustaceans and mollusks are similar to the leukocytes of fish in function and morphology. Type II granulosa cells refer to leukocytes with strong phagocytic ability. Phagocytosis is a method for OsHV-1 to invade cells [49], this may be an essential reason for the susceptibility of type II granulosa cells to OsHV-1 infection. Also, infection with OsHV-1 can induce apoptosis of type II granulosa cells. Some virus particles are encapsulated in apoptotic bodies and destroyed, while others are dissociated in apoptotic host cells illustrating that the apoptosis of type II granulosa cells does not completely eliminate virus particles, and contributing to the mature virus particle diffusion, with cell apoptosis, such that the host immune defenses are weakened. Apoptosis is the main immune escape mechanism of OsHV-1 [50]. Similarly, GCRV may infect specific types of blood cells (leukocytes) through phagocytosis. Studies have shown that GCRV enters the tissue cells mainly through pinocytosis, and then replicating new viral mRNA and translating it into viral protein in the cells to assemble new viral particles [21]. Meanwhile, the structure of virus particles encapsulated in phagocytic vesicles in leukocytes may be like apoptotic bodies, GCRV virus particles are in the process of being destroyed and cleared by leukocytes. Transmission electron microscopy observed that the structure of many leukocytes collapsed and the lesions were serious. There were many empty vesicles inside the leukocytes, which might be in the process of apoptosis caused by GCRV erosion. However, erythrocytes and thrombocytes have complete structures without obvious lesions, which indirectly indicates that GCRV will not infect and damage these cells.

Most non-mammalian erythrocytes, including fish erythrocytes, have nuclei and contain organelles, thus may act as hosts for viral replication [51]. In this study, by flow cytometry, more than 50% of leukocytes after infection were infected with GCRV. The sensitivity of the flow cytometry analysis and the use of such problems as the specificity of the antibodies, may make the number of leukocytes detected on the high side. But as time changes, the positive rate rise, illustrating the proliferation and latency of the GCRV virus in leukocytes. It is concluded that the pathway of GCRV infecting leukocytes is not only the phagocytosis of leukocytes to heterologous substances but also active entrance and coexistence of GCRV in leukocytes. Leukocytes have nuclei and abundant organelles so that they can be used as the host of virus replication.

We observed that in vitro infection with GCRV induced the expression of *IFN1, Mx2, IL-1β*, and *TNFα*, indicating the initiation of antiviral response. The change of *IFN1* and *Mx2* expression levels indicates that GCRV causes the innate antiviral immune response in leukocytes. *Mx2* is a target gene of interferon [26], and its production suggests that interferon is also produced at the protein level after GCRV infection. The up-regulated expression levels of *IL-1β* and *TNFα* indicate that GCRV can induce an intracellular inflammatory response. Interestingly, we also observed that although no obvious GCRV replication was detected in thrombocytes, the expression of *IFN1* was still raised, indicating that the virus can still stimulate the innate immune response of thrombocytes. Similarly, the previous work reported that although IPNV could not infect the erythrocytes of rainbow trout, it could still increase the expression of *IFN-1*, *PKR* and *Mx* genes and induce the antiviral immune response [52]. Therefore, we speculate that although GCRV does not infect thrombocytes, it can stimulate some receptors on their membrane surface to produce the innate antiviral immune response. The mechanism of innate antiviral immune response induced by GCRV to thrombocytes of grass carp remains to be studied.

## 5. Conclusions

In this study, we established the GCRV infection model in vitro by using primary leukocytes to simulate the infection of GCRV on peripheral blood cells of grass carp. This model is helpful to further study the infection, replication, and release of GCRV, as well as to study the antiviral immune response of grass carp leukocytes. This study concluded that leukocytes are the primary target cells of GCRV infection in grass carp, which is of great significance for understanding further the pathogenic mechanism of GCRV and the prevention and control of bleeding diseases in grass carp.

## Figures and Tables

**Figure 1 viruses-13-00870-f001:**
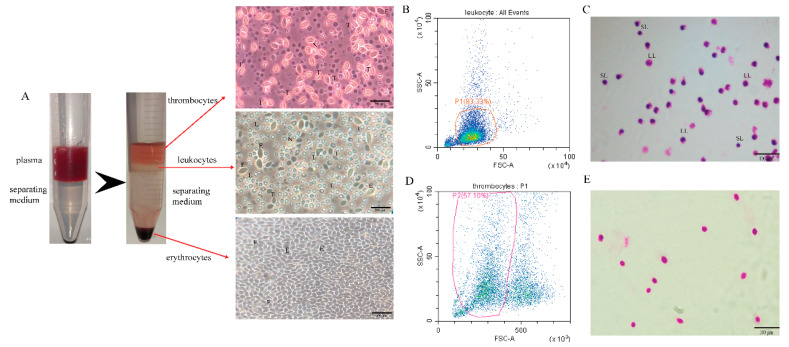
Isolation of leukocytes in grass carp. (**A**) Grass carp peripheral blood cells were subjected to stratification and microscopy examination (scale bar: 200 μm). E: erythrocytes; T: thrombocytes; L: lymphocytes; N: neutrophils. (**B**) Analysis of leukocyte groups by flow cytometry. Group P1 denotes lymphocytes, accounting for about 83%. (**C**) Morphology of lymphocytes with Giemsa staining by light microscopy (scale bar: 100 μm). LL: large lymphocytes; SL: small lymphocytes. (**D**) Analysis of thrombocyte groups by flow cytometry. Group P2 is thrombocytes, accounting for about 57%. (**E**) Morphology of thrombocytes with Giemsa staining by light microscopy (scale bar: 100 μm).

**Figure 2 viruses-13-00870-f002:**
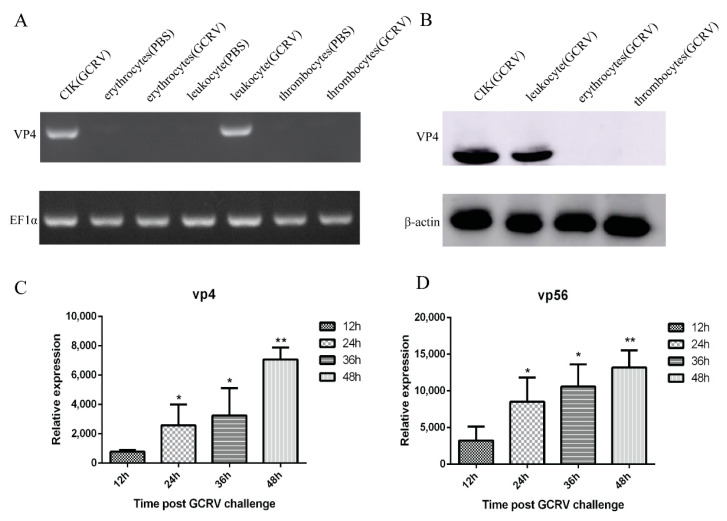
After grass carp reovirus II (GCRV-II) infection, leukocytes, thrombocytes, erythrocytes, and *C. idella* kidney (CIK) cells were detected. (**A**) PCR results showed that leukocytes and CIK cells detected GCRV-positive with VP4 primer and EF1α as an internal reference, but erythrocytes and thrombocytes detected GCRV-negative. (**B**) Western blot results showed that with VP4 polyclonal antibody as primary antibody and β-actin as an internal reference, leukocytes and CIK cells detected GCRV-positive after infection, but erythrocytes and thrombocytes detected GCRV-negative. (**C**,**D**) Quantitative primers of *VP4* and *VP56* were used for qRT-PCR detection, respectively. The viral load of GCRV at 12 h, 24 h, 36 h, and 48 h after GCRV infection was detected, the control group was 0h after GCRV infection. Data of reporter assays and qPCR are shown as mean ± standard deviation (SD) of 6 wells of cell per group and are from one experiment representative of three independent experiments. Significance was calculated in relation to the control group. * *p* < 0.05, ** *p* < 0.01 (two tailed Student’s *t*-tests). The relative transcription levels were normalized to the transcription level of *EF1α* gene and are represented as fold induction relative to the transcription level in control cells, which was set to 1.

**Figure 3 viruses-13-00870-f003:**
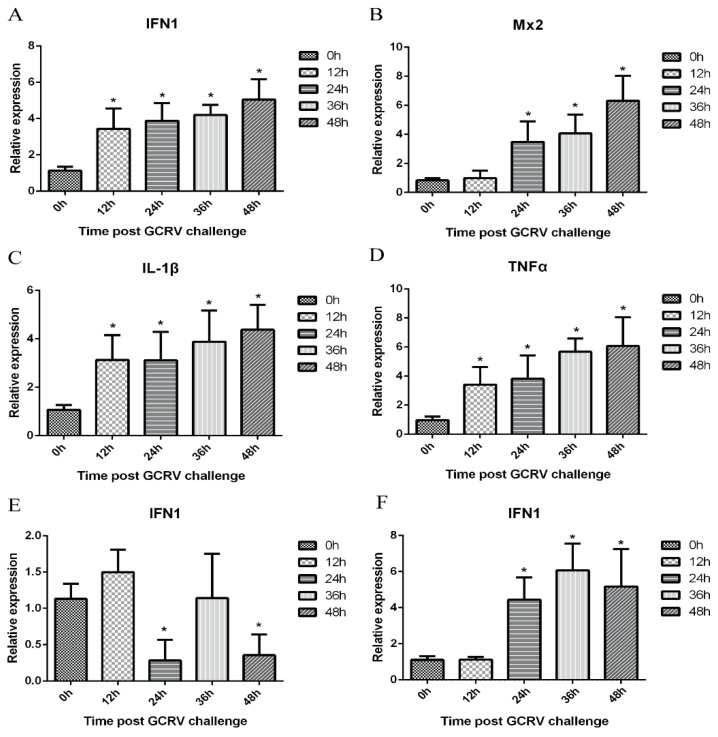
Leukocytic antiviral responses to GCRV-II infection. Expression of genes involved in antiviral responses was measured by qRT-PCR, use *EF1α* as an internal reference gene. The expression levels in infected leukocytes in 6-well plates were challenged with GCRV for 0 h, 12 h, 24 h, 36 h, and 48 h. The relative increase for *IFN1* (**A**), *Mx2* (**B**), *IL-1β* (**C**), and *TNFα* (**D**) is shown. (**E**) The expression levels of *IFN1* in infected erythrocytes in 6-well plates were challenged with GCRV for 0 h, 12 h, 24 h, 36 h, and 48 h. (**F**) The expression levels of *IFN1* in infected thrombocytes in 6-well plates were challenged with GCRV for 0 h, 12 h, 24 h, 36 h, and 48 h. Data of reporter assays and qPCR are shown as mean ± SD of 6 wells of cell per group and are from one experiment representative of three independent experiments. Significance was calculated in relation to the control group. * *p* < 0.05 (two tailed Student’s *t*-tests). The relative transcription levels were normalized to the transcription level of *EF1α* gene and are represented as fold induction relative to the transcription level in control cells, which was set to 1.

**Figure 4 viruses-13-00870-f004:**
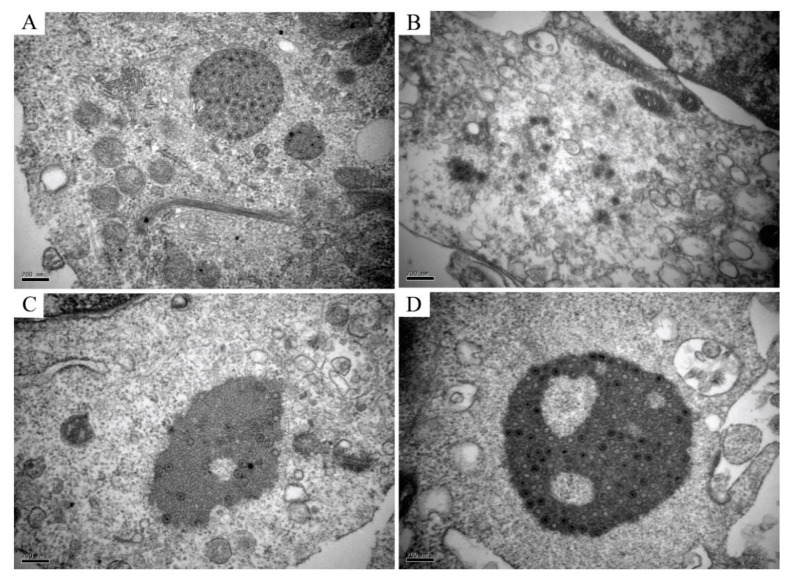
Several different morphogenesis of GCRV-II in CIK cells. (**A**) Mature virions arranged neatly like a lattice, with a diameter of about 70–80 nm; (**B**) Scattered virions, released in the cytoplasm, with a diameter of about 70–80 nm; (**C**) Early virus inclusion bodies are relatively dense and contain a small number of virions, which are in the process of forming; (**D**) Late VIBs, which are thick and have a large number of virions, are in the process of generating.

**Figure 5 viruses-13-00870-f005:**
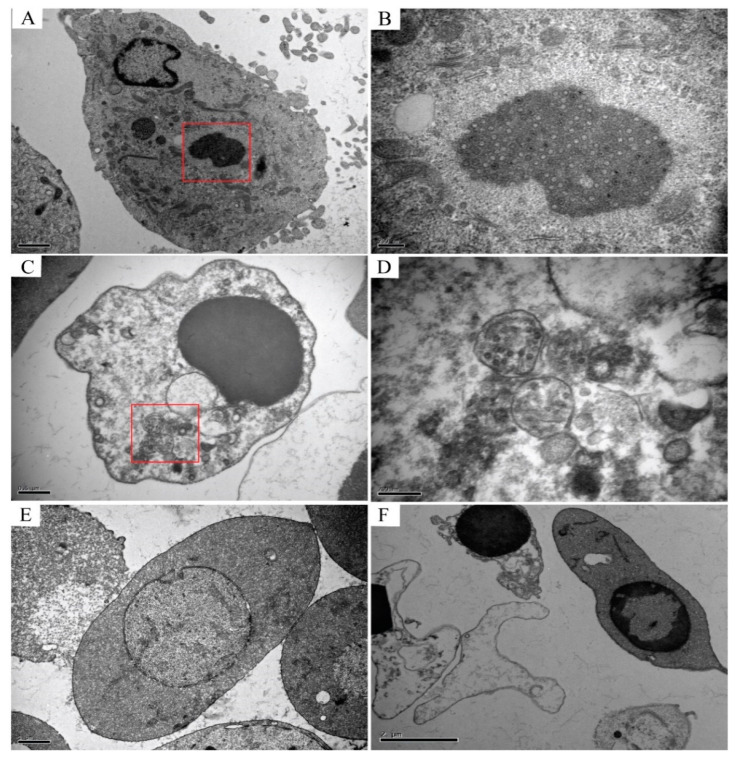
Transmission electron microscope observation of leukocytes, erythrocytes, and thrombocytes after infection with GCRV-II virus. (**A**,**B**) are the virus inclusion bodies (VIBs) in CIK cells infected with the virus. The structure is dense and contains a large number of virions in the process of being formed, without apparent membrane structure outside; (**C**,**D**) are virions in infected leukocytes that exist in the cytoplasm in the form of a relatively obvious envelope structure, which envelops a large number of virions, and the interior is not dense, showing a vacuole structure; (**E**) and (**F**) are respectively infected erythrocytes and thrombocytes, both cells have no noticeable pathological changes, and their constructions are complete.

**Figure 6 viruses-13-00870-f006:**
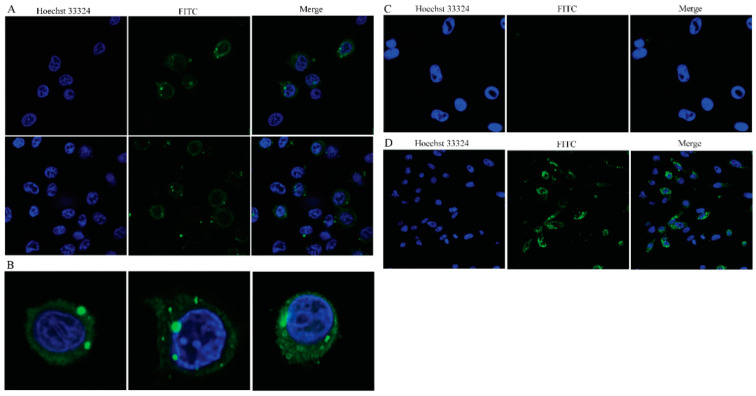
Immunofluorescence of GCRV-II infected peripheral blood cells. (**A**) Fluorescent labeling of the GCRV-II vp4-protein (green) in ex vivo infected leukocytes, the nuclei were stained with Hoechst (blue). Confocal microscopy of different staining patterns in the cytoplasm, including a few inclusions and scattered granular staining in the perinuclear region (scale bar: 10 μm). (**B**) Confocal microscopy images showing viral inclusions in the perinuclear area. (**C**) Fluorescent labeling of the GCRV-II vp4-protein (green) in ex vivo infected erythrocytes, the nuclei were stained with Hoechst (blue). Confocal microscopy images showing erythrocytes were GCRV negative (scale bar: 10 μm). (**D**) Fluorescent labeling of the GCRV-II vp4-protein (green) in infected CIK cells, the nuclei were stained with Hoechst (blue). Confocal microscopy images showing many virus inclusions were found in the cytoplasm (scale bar: 20 μm).

**Figure 7 viruses-13-00870-f007:**
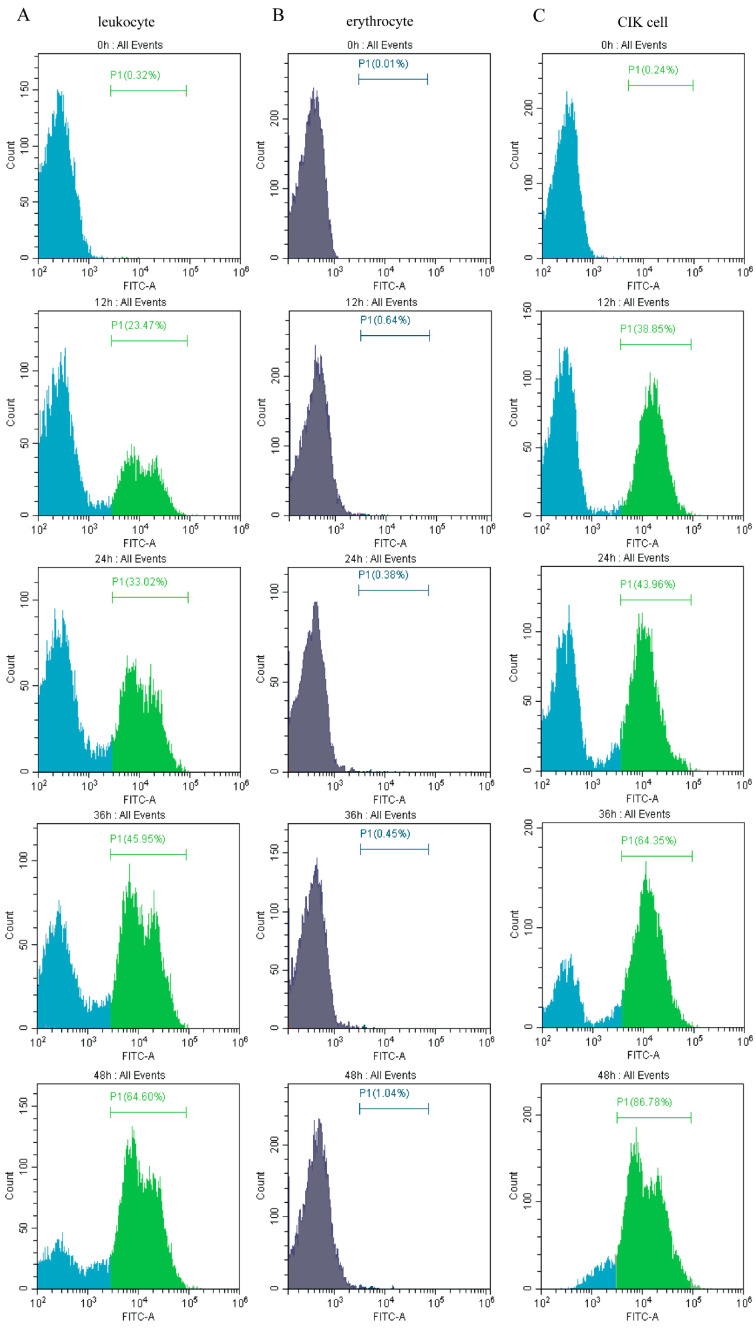
GCRV-positive leukocytes, GCRV-negative erythrocytes and GCRV-positive CIK cells detected by flow cytometry. Flow cytometry result from the intracellular staining for GCRV vp4 protein from GCRV infected cell (green) and control cell (blue) at 0 h, 12 h, 24 h, 36 h and 48 h post infection. The fluorescence intensity (GCRV vp4) is shown on the X-axis and cell count on the y-axis counting 10,000 cells per sample. (**A**) In ex vivo infected leukocytes, GCRV-positive rates were 0.32%, 23.47%, 33.02%, 45.95%, 64.60% at 0 h, 12 h, 24 h, 36 h and 48 h, respectively. (**B**) In ex vivo infected erythrocytes, GCRV-positive rates were 0.01%, 0.64%, 0.38%, 0.45%, 1.04% at 0h, 12h, 24h, 36h and 48h, respectively. (**C**) In infected CIK cells, GCRV-positive rates were 0.24%, 38.85%, 43.96%, 64.35%, 86.78% at 0 h, 12 h, 24 h, 36 h and 48 h, respectively.

**Table 1 viruses-13-00870-t001:** Primer sequences used for quantitative real-time polymerase chain reaction (qRT-PCR) analysis in this study.

Gene Name	Primer Name	Sequence (5′→3′)	Accession Number
IFN1	IF590	AAGCAACGAGTCTTTGAGCCT	DQ357216.1
	IR591a	CGCTCAATCTTCCATCCCTT	
EF1α	EF125	CGCCAGTGTTGCCTTCGT	GQ266394
	ER126	CGCTCAATCTTCCATCCCTT	
Mx2	MF428	ACATTGACATCGCCACCACT	JF699168.1
	MF429	TTCTGACCACCGTCTCCTCC	
TNFα	TnfF169	GCTGCTGTCTGCTTCACGC	HQ696609.1
	TnfR170	AGCCTGGTCCTGGTTCACTCT	
IL-1β	IL-1βF663	CAGTGCTCCATTTGTGATCAG	MK942107.1
	IL-1βR664	GAAATGGCCAGACACACAGG	
VP4	VF146	CGAAAACCTACCAGTGGATAATG	MN136091
	VR147	CCAGCTAATACGCCAACGAC	
VP56	VF73	AGCAGGCTATTCATCACCAGT	MK675081
	VR74	GTTCTAACGCTCACCGTCTTTTC	

## Data Availability

Not applicable.

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
