# Peer review of "Type II Grass Carp Reovirus Infects Leukocytes but Not Erythrocytes and Thrombocytes in Grass Carp (*Ctenopharyngodon idella*)"

_viruses, 2021, doi:10.3390/v13050870_

Round 1
Reviewer 1 Report
This article is valuable for the study of fish reoviruses, describes several methods, which were properly used in research and many well-described Figures enrich the work.
Extensive "Introduction" influence on better understanding problem of grass carp reovirus infection , but there is no explanataion of meaning CIK cells in "Introduction" part or in "Keywords". I recommend to explain this shortcut to better understand the text.
"Materials and Methods"
2.2 - There is no information about fish abundance, how many times were fish caught to take blood samples and from how many fish.
2.9 - There is no information about how you obtained primer sequences, from other publication or by using some kind of software?
2.10 - The statistical analysis is mentioned but there is no elaboration in the "Results" part .
Line 226 - you mentioned about β-actin as the reference, but there is no previous information about it
Reviewer 2 Report
General comments
In the field of viral diseases of fish, the investigation of blood cells has been frequently overlooked. Thus, this paper by Yang and cols. is a potential valuable contribution to the knowledge of fish viruses pathogenesis, particularly in carp. In this work, peripheral blood cells from carp are separated into erythrocytes, thrombocytes and leukocytes to be infected with GCRV “ex vivo”. Virus replication within those cells was measured and compared to a carp kidney cell line. Ex vivo infections have been widely accepted as good models of viral infection in fish. Overall, I believe the work has good quality and present interesting novel data. My major issue would be the absence of GCRV data from the real-time PCR measurements of GCRV-infected cell samples. Some results, like the fluctuating levels on ifn1 in GCRV-infected erythrocytes (Fig.3 E) are difficult to make sense of without a parallel determination of viral RNA levels. Moreover, it remains unclear whether the replication of GCRV in either erythrocytes or thrombocytes might have been detected by qPCR.
Minor issues
The Introduction seems to be lacking some information on the effect of GCRV on leukocytes, erythrocytes and thrombocytes in the infected carp. Is there anything in the literature on those cells (like cell count) in GCRV-infected fish?
Was qRT-PCR detection of GCRV in erythrocytes or in thrombocytes attempted? If so, it so this should be mentioned in 3.2., even if the results were negative.
Figure 2, panels C and D: what cells were used in the experiment?
With respect to the immune response after GCRV infection (figure 3) I wonder why the gig1/gig2 genes that were discovered specifically associated with carp infection with GCRV (Sun, 2013) were not analyzed.
How is GCRV replication in leukocytes compared to CIK cells? Some data on virus yield (titers) in both cell types may be helpful to determine how efficient is GCRV growth in leukocytes.
The presence of those enveloped vacuole-like structures enclosing virions in the cytoplasm of the GCRV-infected leukocytes (figure 5) is a very interesting finding. Do the authors have any clue on where those membranes are originated from? (ER? Golgi?).
Some citations of previous works on fish blood cells infected with dsRNA viruses are missing in this paper (see below). For instance, the following article on infection of fish erythrocytes with another dsRNA virus (IPNV), showing stimulation of the innate immune response in spite of non apparent viral replication.
Nombela, I., Carrion, A., Puente-Marin, S., Chico, V., Mercado, L., Perez, L., Coll, J. and Ortega-Villaizan, M. 2017. Infectious pancreatic necrosis virus triggers antiviral immune response in rainbow trout red blood cells, despite not being infective
Rosaeg, M.V., Lund, M., Nyman, I.B., Markussen, T., Asperhaug, V., Sindre, H., Dahle, M.K. and Rimstad, E. 2017. Immunological interactions between Piscine orthoreovirus and Salmonid alphavirus infections in Atlantic salmon.
Sun, C., Liu, Y., Hu, Y., Fan, Q., Yu, X., Mao, H. and Hu, C. 2013. Gig1 and Gig2 homologs (CiGig1 and CiGig2) from grass carp (Ctenopharyngodon idella) display good antiviral activities in an IFN-independent pathway.
